# Recruitment of ubiquitin-activating enzyme UBA1 to DNA by poly(ADP-ribose) promotes ATR signalling

Ramhari Kumbhar[1], Sophie Vidal-Eychenié[1], Dimitrios-Georgios Kontopoulos[2], Marion Larroque[3], Christian Larroque[4], Jihane Basbous[1], Sofia Kossida[1,5], Cyril Ribeyre[1], Angelos Constantinou[1]

**The DNA damage response (DDR) ensures cellular adaptation to genotoxic insults. In the crowded environment of the nucleus, the assembly of productive DDR complexes requires multiple protein modifications. How the apical E1 ubiquitin activation enzyme UBA1 integrates spatially and temporally in the DDR remains elusive. Using a human cell-free system, we show that poly(ADP-ribose) polymerase 1 promotes the recruitment of UBA1 to DNA. We find that the association of UBA1 with poly(ADP-ribosyl)ated protein–DNA complexes is necessary for the phosphorylation replication protein A and checkpoint kinase 1 by the serine/threonine protein kinase ataxia-telangiectasia and RAD3-related, a prototypal response to DNA damage. UBA1 interacts directly with poly(ADP-ribose) via a solvent-accessible and positively charged patch conserved in the Animalia kingdom but not in Fungi. Thus, ubiquitin activation can anchor to poly(ADP-ribose)-seeded protein assemblies, ensuring the formation of functional ataxia-telangiectasia mutated and RAD3-related-signalling complexes.**

## Introduction

The DNA damage response (DDR) is a signal transduction pathway that detects lesions in DNA and ensures cell and organismal survival through coordination of DNA repair and DNA replication with physiological processes, including cell cycle progression and transcription (Matsuoka et al, 2007; Ciccia & Elledge, 2010). At the apex of the DDR, the master checkpoint kinases ataxia telangiectasia mutated (ATM) and ataxia-telangiectasia and RAD3-related (ATR) and the poly(ADP-ribose) (pADPr) polymerases (PARP1) sense and signal double-strand DNA (dsDNA) breaks (DSBs) and the slowing or stalling of replication forks (Ciccia & Elledge, 2010; Blackford & Jackson, 2017; Ray Chaudhuri and Nussenzweig, 2017;

Saldivar et al, 2017). Induction of the DDR triggers a cascade of protein modifications by ADP-ribosylation, phosphorylation, SUMOylation, ubiquitylation, acetylation, and methylation, which collectively promote the assembly of DNA damage signalling and DNA repair proteins into discrete chromatin foci (Ciccia & Elledge, 2010; Dantuma & van Attikum, 2016).

One of the earliest responses to DNA damage is the conjugation by PARP1 of pADPr to substrate proteins, including itself, at DNA breaks and stalled replication forks (Caldecott et al, 1996; Bryant et al, 2009; Langelier et al, 2011). PARP1 activity is induced by discontinuous DNA structures such as nicks, DSBs, and DNA cruciform (Caldecott et al, 1996; Bryant et al, 2009; Langelier et al, 2011). The negatively charged pADPr polymers recruits a large spectrum of proteins (Gagne et al, 2008, 2012), including FET (FUS [fused in liposarcoma], EWS [Ewing sarcoma] and TAF15 [TATA binding associated factor 15]) family proteins FUS and TAF15 that rapidly accumulate at DNA lesions induced by micro-irradiation (Altmeyer et al, 2015; Izhar et al, 2015; Patel et al, 2015). Upon reaching a critical concentration, FET family proteins phase separate into liquid droplets under physiological conditions (Altmeyer et al, 2015; Patel et al, 2015). Hyperactivation of PARP1 at DNA breaks seeds liquid phase separation (Altmeyer et al, 2015; Patel et al, 2015). The rapid recruitment of FUS and TAF15 at DNA damage sites is followed by their prolonged exclusion in a manner that depends on the kinase activity of ATM, ATR, and DNA-dependent protein kinase catalytic subunit (DNA-PKcs) (Britton et al, 2014). Phosphorylation of FUS at multiple consensus serine/threonine glutamine sites by DNA-PKcs counteracts the self-association and aggregation of its low-complexity domain (Monahan et al, 2017).

Protein ubiquitylation is extensive at sites of DNA damage (Meerang et al, 2011; Lee et al, 2017; Baranes-Bachar et al, 2018). Yet, it is unclear how the E1–E3 enzymatic cascade is organised in space and time to permit high fluxes of ubiquitin attachment to protein substrates at sites of DNA damage. Ubiquitin is first adenylated by an E1 ubiquitin-activating enzyme, transferred to a carrier E2

[1]Institut de Génétique Humaine, Centre National de la Recherche Scientifique, Université de Montpellier, Montpellier, France   [2]Imperial College London, Department of Life Sciences, Silwood Park Campus, Ascot, UK   [3]Institut du Cancer de Montpellier, Montpellier, France   [4]Institut de Recherche en Cancérologie de Montpellier, Université de Montpellier, Institut National de la Santé et de la Recherche Médicale, Montpellier, France   [5]IMGT, The International ImMunoGeneTics Information System, Montpellier, France

Cyril Ribeyre and Angelos Constantinou jointly supervised this work.
Correspondence: angelos.constantinou@igh.cnrs.fr
Ramhari Kumbhar's present address is Department of Molecular Biosciences, Institute for Cellular and Molecular Biology, The University of Texas at Austin, Austin, TX, USA.

ubiquitin-conjugating enzyme in preparation for the recognition by an E3 ubiquitin ligase of the target ubiquitylation substrate. UBA1 is the E1 enzyme at the apex of ubiquitin signalling in the DDR (Moudry et al, 2012).

The master checkpoint kinase ATR is activated by a fail-safe multistep mechanism involving the recruitment of sensor and mediator proteins at stalled replication forks or resected DNA ends (Marechal & Zou, 2013; Saldivar et al, 2017). ATR in turn activates its major substrate effector checkpoint kinase 1 (Chk1) (Guo et al, 2000; Hekmat-Nejad et al, 2000; Liu et al, 2000; Zhao & Piwnica-Worms, 2001). DNA replication stress, defined as the slowing or stalling of replication forks, typically yields 70- to 500-nucleotide long stretches of single-stranded DNA (ssDNA) (Sogo et al, 2002; Hashimoto et al, 2010; Zellweger et al, 2015). In addition to ssDNA, a 5′-ended ssDNA–dsDNA junction is required for ATR activation in *Xenopus laevis* egg protein extract (MacDougall et al, 2007). Replication protein A (RPA)-covered ssDNA recruits and increases the local concentration of ataxia telangiectasia mutated and Rad3-related interacting protein (ATRIP)-ATR at DNA damage sites (Zou & Elledge, 2003). RPA also interacts with NBS1 (Shiotani et al, 2013). The MRE11–RAD50–NBS1 complex recruits DNA topoisomerase 2-binding protein 1 (TOPBP1) to the ATR-activating DNA structures (Duursma et al, 2013; Shiotani et al, 2013). RAD17 loads the RAD9-RAD1-HUS1 (9-1-1) clamp at ds/ssDNA junctions (Zou et al, 2002; Ellison & Stillman, 2003). TOPBP1 and the 9-1-1 clamp activate ATR (Kumagai et al, 2006; Delacroix et al, 2007; Mordes et al, 2008; Yan & Michael, 2009; Duursma et al, 2013), which in turn phosphorylates and activates effector Chk1. Full ATR activation also requires RHINO (RAD9, RAD1, HUS1-interacting nuclear orphan protein), a protein that binds independently to TOPBP1 and the 9-1-1 complex (Cotta-Ramusino et al, 2011).

ATR is also activated by DNA structures that contain DNA ends. In *X. laevis* egg extracts, the homopolymer poly(dA)$_{70}$-poly(dT)$_{70}$ triggers activation of ATR independently of RPA through phosphorylation of TOPBP1 by ATM on serine 1131 (Yoo et al, 2007). The key structural features of poly(dA)$_{70}$-poly(dT)$_{70}$ that induce ATR activation, however, remain ill defined. Linear dsDNA substrate can induce ATR activation through progressive resection of the DNA ends (Shiotani & Zou, 2009). Alternatively, we reported earlier that a linear DNA duplex bearing an internal ssDNA gap promptly activates endogenous ATR in human cell extracts (Vidal-Eychenie et al, 2013). In this system, the key structural feature of the ATR-activating DNA substrate is the juxtaposition of an accessible DNA end and a short ssDNA gap, which is both sufficient to trigger ATR signalling and permissive to variations of the distance between the DNA ends and the ssDNA region (Vidal-Eychenie et al, 2013). In protein extracts, which partially recapitulate the complexity of the nuclear interior, the activity of DNA-PKcs promotes the assembly of a functional ATR-signalling complex on gapped linear duplex DNA (Vidal-Eychenie et al, 2013).

In addition to the canonical ATR activation pathway, accumulating evidence suggests that cells are endowed with distinct ATR activation systems. ATR is also activated by Ewing's tumour-associated antigen 1 independently of TOPBP1 at stalled replication forks (Bass et al, 2016; Haahr et al, 2016; Lee et al, 2016). Furthermore, ATR can be activated at the nuclear envelope in response to mechanical stress, independently of RPA, TOPBP1, and the 9-1-1 complex (Kumar et al, 2014).

Protein phosphorylation is just one among diverse posttranslational modifications that are required for the correct execution of the ATR-Chk1 signalling pathway. Chk1 includes a pADPr—binding motif and associates with pADPr chains independently of ATR (Min et al, 2013). Chk1 binding to pADPr is necessary for Chk1 localisation and activation near replication forks (Min et al, 2013). ATRIP SUMOylation acts as a "glue," promoting ATR signalling via stimulation of ATRIP association with ATR, RPA, TOPBP1, and the MRE11-RAD50-Nbs1 complex (Wu et al, 2014). Furthermore, RPA ubiquitylation by the E3 ubiquitin ligases PRP19 and RFW3 promotes ATR activation (Gong & Chen, 2011; Marechal et al, 2014; Elia et al, 2015).

Here, we used human cell extracts to explore how the apical posttranslational modification enzymes PARP1, UBA1, and ATR are coordinated, using ATR signalling as an archetypal DDR. Linear DNA substrates promptly induced protein poly(ADP-ribosyl)ation, ubiquitylation, and phosphorylation in human protein extracts. We detected UBA1 among DNA-bound proteins. The synthesis of pADPr chains by PARP1 promoted UBA1 recruitment to DNA, ubiquitylation of DNA-bound proteins, and phosphorylation of the ATR substrate proteins RPA and Chk1. We observed that human UBA1 has affinity for pADPr polymers. By contrast, *Saccharomyces cerevisiae* UBA1 did not bind to pADPr, consistent with the fact that fungi do not possess a poly (ADP-ribosyl)ation system. We exploited this difference between Animalia and yeast UBA1 to identify, via phylogenetic analyses and site-directed mutagenesis, a positively charged patch that endows Animalia UBA1 with the capacity to associate with pADPr. We discuss how PARP1-coupled UBA1 activity may ensure the functionality of pADPr-seeded protein assemblies, as illustrated here with the robust and rapid activation of endogenous ATR in a complex protein mixture.

## Results

### UBA1 functionally associates with an ATR-activating DNA structure

A singularly active ATR-signalling complex self-assembles in human protein extracts upon incubation with a gapped linear duplex DNA (Vidal-Eychenie et al, 2013) (Fig 1A–C). To gain further insights into the mechanism of assembly and activation of DNA damage signalling complexes, we biotinylated the linear duplex DNA structures to perform pull-down experiments. After incubation for 10 min at 37°C in human nuclear extracts, we captured the biotinylated DNA structure with streptavidin-coated beads and resolved DNA-bound proteins by PAGE (Fig 1A and B). Silver staining revealed defined protein bands that were enriched in DNA pull-downs (Fig 1B). The spectrum of proteins pulled down with gapped linear duplex DNA was different from that of linear duplex DNA (Fig 1B), confirming that the two DNA substrates are different. We excised and analysed gapped DNA–associated proteins by mass spectrometry. Among the most prominent DNA-bound proteins, we identified DNA-PKcs, PARP1, UBA1, as well as RNA-processing proteins such as FUS and HNRNPUL1 (Fig 1B). Although silver staining revealed a different spectrum of proteins associated with linear duplex and gapped linear DNA substrates, DNA-PKcs, PARP1, and UBA1 bound indiscriminately to both DNA substrates, as revealed by Western blotting (Fig 1C). DNA-PK and PARP1 are recruited to DNA ends, which are

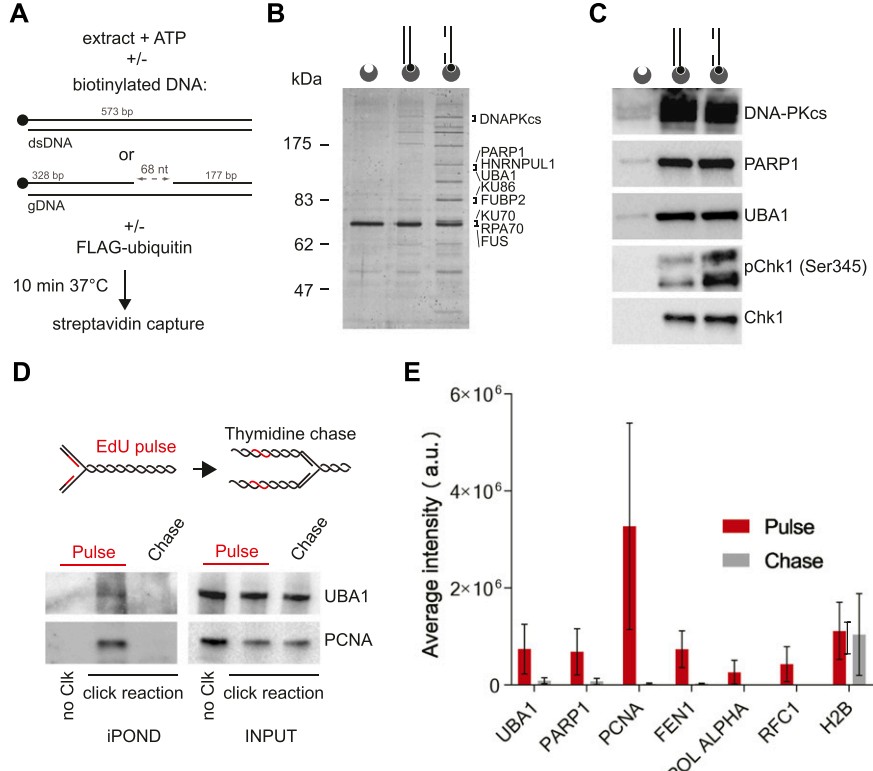

**Figure 1. UBA1 is recruited to an ATR-activating structure and is associated with ongoing replication forks.**
**(A)** Experimental scheme: Biotinylated duplex DNA structures are incubated with nuclear extract in the presence of ATP. After 10 min at 37°C, DNA-bound proteins are pulled down using streptavidin-coated beads. To monitor protein ubiquitylation in vitro, reaction mixtures are supplemented with recombinant FLAG-ubiquitin. **(B)** Nuclear extracts were incubated with the indicated biotinylated DNA substrates. DNA-bound proteins were isolated, resolved by PAGE, stained with silver, and identified by mass spectrometry. The most abundant proteins are indicated. **(C)** Indicated proteins were pulled down with streptavidin-coated beads coupled to duplex DNA or gapped duplex DNA and detected by Western blotting. The duplex DNA and gapped duplex DNA substrates are represented schematically. Biotin (black circles) and streptavidin-coated beads (dented grey circles) are shown. **(D)** Indicated proteins were isolated by iPOND and detected by Western blotting. HEK293 cells were pulse-labelled with 5-ethynyl-2′-deoxyuridine (EdU) for 10 min and chased with thymidine for 60 min. In no click samples (no Clk), desthiobiotin-TEG azide was replaced with DMSO. **(E)** Bar plot showing average peptides intensities (MaxQuant label-free quantification) corresponding to the indicated proteins. HeLa S3 cells were pulse-labelled with EdU for 5 min and chased with thymidine for 120 min. Pulse experiment has been repeated three times and chase experiment two times. Error bars represent the standard variation. The UBA1 peptides identified are listed in Table S1.

common to both DNA substrates. Consistent with previous observations (Vidal-Eychenie et al, 2013), the presence of a single-strand DNA gap in duplex DNA stimulated the phosphorylation of the ATR substrate protein Chk1 on Serine 345 (Fig 1C). Chk1 was also phosphorylated, albeit less efficiently, in reaction mixtures containing linear duplex DNA (Fig 1C). In these electrophoretic conditions, we observed two phospho-Chk1 (Ser345) signals with distinct mobility that could reflect additional Chk1 modifications.

The presence of UBA1 in DNA pull-downs was intriguing because it is unclear whether UBA1 is recruited to chromatin at DNA damage sites (Moudry et al, 2012; Izhar et al, 2015). In one study, UBA1 tagged with GFP was not detectable at DNA damage sites (Moudry et al, 2012), but a systematic analysis revealed UBA1 among a list of about 120 proteins recruited to DNA breaks marked by γH2AX co-staining (Izhar et al, 2015). We confirmed the binding of UBA1 to DNA in vitro using an anti-FLAG antibody and nuclear extracts prepared from cells expressing FLAG-tagged UBA1 (Fig S1A). We noticed that UBA1 was among proteins identified by mass spectrometry in the vicinity of replication forks using isolation of proteins on nascent DNA (iPOND) (Lossaint et al, 2013; Ribeyre et al, 2016). In this procedure, the newly synthesised DNA is labelled with the thymidine analogue EdU before formaldehyde cross-linking and protein capture on EdU-labelled DNA (Sirbu et al, 2013). This suggests that UBA1 can localise to chromatin in living cells, at least transiently near replication forks. To confirm the presence of UBA1 near replication forks, we probed UBA1 by immunoblotting in EdU pull-downs from HEK293 cells. We detected a signal for UBA1 on nascent DNA and this signal was lost after thymidine chase (Fig 1D), indicating that UBA1 accumulates in

the vicinity of replication forks. To confirm this observation further and provide a highly specific alternative to immunoblotting, we performed liquid chromatography tandem-mass spectrometry analyses using a TripleTOF system and used MaxQuant for label-free quantification (Cox & Mann, 2008). We identified UBA1 on nascent DNA (Fig 1E). The UBA1 peptides detected in pulse-chase experiments are indicated in Table S1. Calculation of the sums of all individual peptide intensities revealed that the relative amount of UBA1 on nascent DNA drops dramatically after thymidine chase. The level of detection and the dynamics of UBA1 on nascent DNA were similar to that of PARP1, PCNA, FEN1, polymerase α, and RFC1 (Fig 1E). The data indicate that UBA1 is transiently recruited to chromatin near replication forks.

To verify that UBA1 and the whole ubiquitylation cascade is active during incubation of human protein extracts with duplex DNA structures, we supplemented the reaction mixture with FLAG-tagged ubiquitin. Anti-FLAG immunoblotting of proteins bound to biotinylated DNA structures revealed an intense smearing signal resulting from the conjugation of ubiquitin to proteins (Figs 2A and S1B). A small inhibitor of UBA1, PYR41 (Yang et al, 2007), abolished the conjugation of DNA-bound proteins to ubiquitin (Fig 2A). To confirm that UBA1 is the E1 enzyme involved in the ubiquitylation of proteins induced by linear duplex DNA structures, we prepared nuclear extracts from cells transfected either with a control siRNA or with a siRNA against UBA1 (Fig 2B). Ubiquitylation of DNA-bound proteins was strictly dependent on the E1 enzyme UBA1 (Fig 2C).

Next, we examined the induction of Chk1 phosphorylation by defined DNA substrates to investigate if UBA1 activity is coordinated

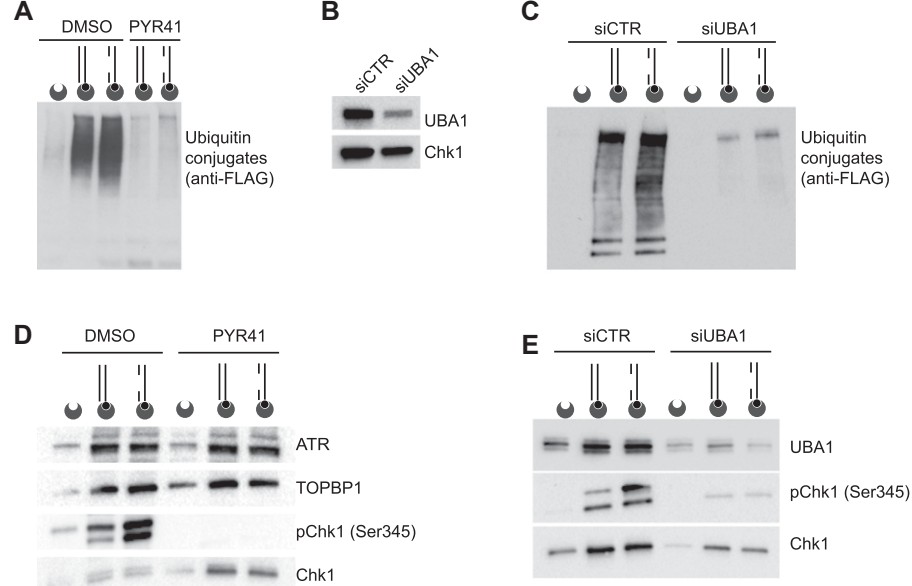

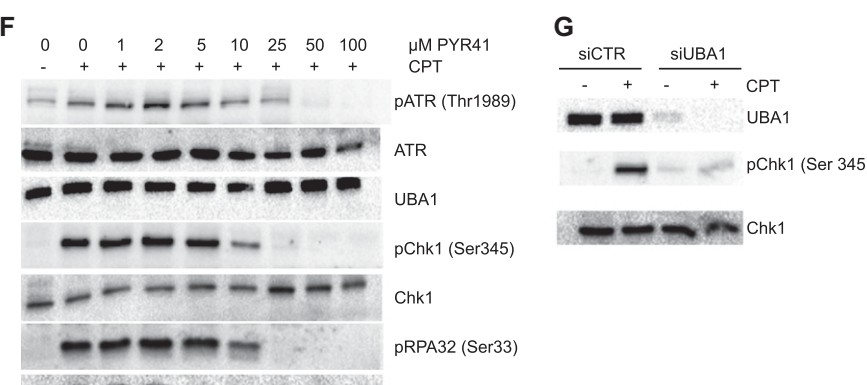

**Figure 2. UBA1 promotes Chk1 activation.**
**(A)** DNA-bound proteins conjugated to FLAG-ubiquitin were isolated from nuclear extracts supplemented with solvent (DMSO) or PYR41 and immunoblotted with anti-FLAG antibody. The DNA substrates are represented schematically. Biotin (black circles) and streptavidin-coated beads (dented grey circles) are shown. **(B)** Immunoblotting of UBA1 and Chk1 in nuclear extracts prepared from HeLa S3 cells treated with siRNAs against UBA1 (siUBA1) or control (siCTR). **(C)** Ubiquitylated proteins bound to DNA were isolated from control and UBA1 knockdown cells and analysed as described in (A). **(D)** Western blot analysis of the indicated proteins isolated with DNA structures from protein extracts supplemented with PYR41 or solvent (DMSO). **(E)** DNA-bound proteins were pulled down from control or anti-UBA1 siRNA nuclear extracts and analysed by Western blotting as indicated. **(F)** HeLa S3 cells pretreated or not with increasing doses of PYR41 were exposed to 1 μM CPT for 120 min and the indicated proteins were probed by Western blotting. **(G)** HeLa S3 cells treated with siRNAs against UBA1 (siUBA1) or control (siCTR) cells were treated or not with 1 μM CPT for 120 min and probed for the indicated proteins by Western blotting.

with ATR signalling in this experimental system. In the presence of the solvent DMSO, gapped linear duplex DNA strongly induced Chk1 phosphorylation, whereas phosphorylation of the ATR substrate was completely blocked by the UBA1 inhibitor PYR41 (Fig 2D). Intriguingly, PYR41 induced the accumulation of un-phosphorylated Chk1 on the biotinylated DNA structures, suggesting that PYR41 induces Chk1 aggregation on DNA (Fig 2D). Suppression of UBA1 by means of siRNAs also inhibited Chk1 phosphorylation (Fig 2E). Interestingly, depletion of UBA1 preferentially suppressed the Chk1 phospho-signal of slower mobility, suggesting that this signal could correspond to ubiquitinated phospho Chk1 (Fig 2E).

To verify the consequence of UBA1 inhibition by PYR41 on ATR signalling in cells, we induced activation of cellular ATR using camptothecin (CPT) and methyl methanesulfonate. Consistent with data obtained using the cell-free system, addition of increasing concentrations of PYR41 to the cell culture medium of CPT-treated cells progressively blocked ATR signalling, as revealed by inhibition of ATR auto-phosphorylation on Thr1989, of RPA32 phosphorylation

on Ser 33, and of Chk1 phosphorylation on Ser 345 (Fig 2F). Consistent with this, UBA1 depletion by means of siRNA inhibited CPT-induced Chk1 phosphorylation (Fig 2G). PYR41 also inhibited the induction of Chk1 phosphorylation by methyl methanesulfonate treatment (Fig S1C). Collectively, these data indicate that UBA1 activity is required for ATR signalling.

## Poly(ADP-ribosyl)ation mediates UBA1 recruitment to DNA

The analysis of proteins bound to the substrate by mass spectrometry and Western blot revealed the presence of PARP1 (Fig 1B–C), which binds to and is activated by discontinuous DNA structures (Gibson & Kraus, 2012). As poly(ADP-ribosyl)ation is involved in the recruitment of a high diversity of proteins in the DDR (Gagne et al, 2008, 2012), we examined if DNA-induced protein ubiquitylation in human protein extracts was dependent on PARP1 activity. First, we tested if poly(ADP-ribosyl)ation is occurring during the course of the in vitro reaction. For this purpose, we supplemented reaction mixtures containing human protein extracts and

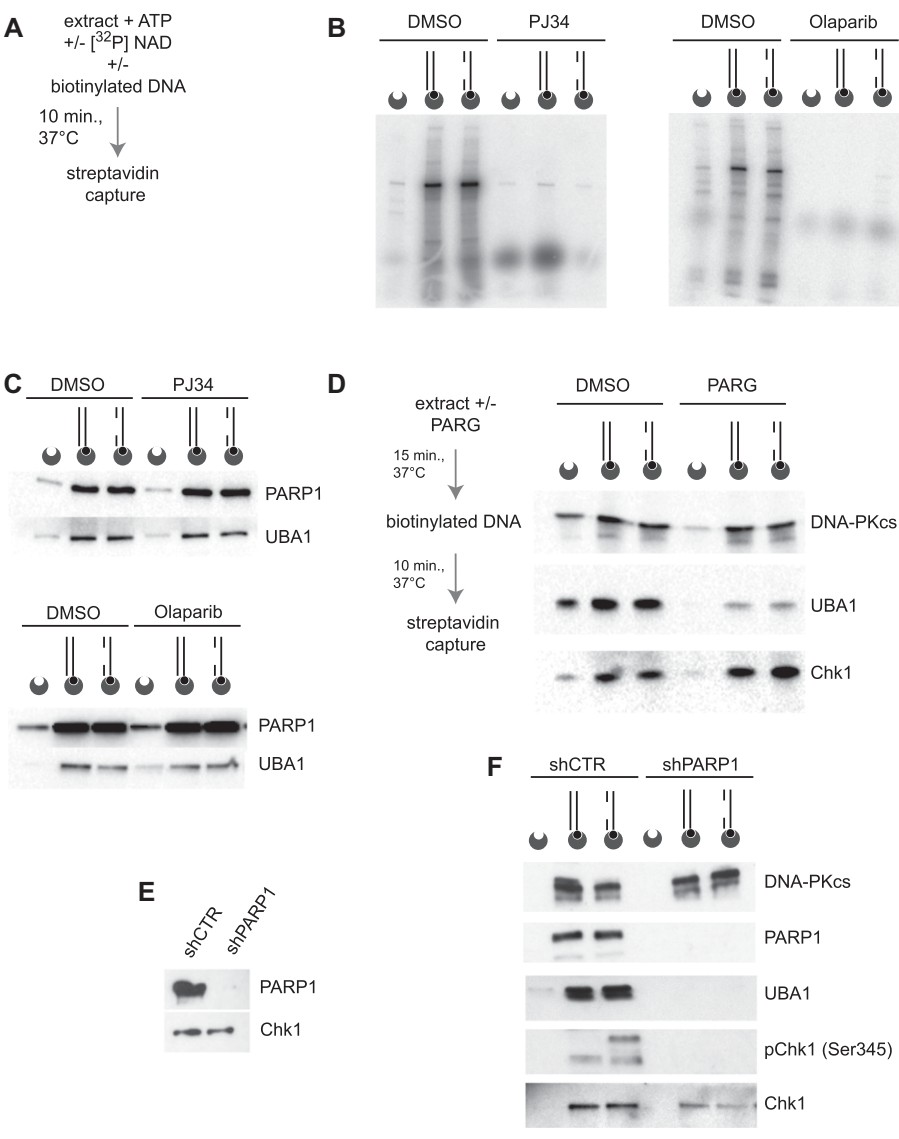

**Figure 3.  pADPr chains are required for UBA1 recruitment.**
**(A)** Reaction mixtures were incubated with [$^{32}$P]-labelled NAD, and DNA-bound proteins were pulled down and resolved by PAGE. Poly(ADP-ribosyl)ated proteins were revealed by autoradiography. **(B)** Autoradiography of proteins poly(ADP-ribosyl)ated in nuclear extracts and isolated with the indicated DNA substrates represented schematically. Biotin (black circles) and streptavidin-coated beads (dented grey circles) are shown. When indicated, reactions were performed in the presence of the PARP1 inhibitors PJ34 or olaparib. **(C)** Western blot analysis of the indicated proteins isolated with biotinylated DNA substrates from reaction mixtures supplemented, when indicated, with PJ34 or olaparib. **(D)** Nuclear extracts were pretreated with PARG when indicated and incubated with DNA substrates. The indicated proteins were isolated by DNA pull-down and revealed by Western blotting. **(E)** Western blot analysis of PARP1 and Chk1 in nuclear extracts prepared from cells treated with anti-luciferase (shCTR) or anti-PARP1 (shPARP1) shRNAs. **(F)** The indicated proteins were isolated along with biotinylated DNA substrates after incubation in protein extracts from cells treated with control (shCTR) or anti-PARP1 shRNAs, as indicated.

linear duplex DNA with [$^{32}$P]-labelled nicotinamide adenine dinucleotide (NAD+) precursor. After incubation for 10 min at 37°C, we used streptavidin-coated beads to retrieve the biotinylated DNA from the reaction mixture, resolved DNA-bound proteins by PAGE, and revealed pADPr signals by autoradiography (Fig 3A and B). Electrophoretic resolution of pull-downs from linear duplex and gapped linear duplex DNA yielded similar poly(ADP-ribosyl)ated protein signals (Fig 3B), indicating that endogenous PARP1 is activated by linear duplex DNA structures in protein extracts. A prominent [$^{32}$P]-labelled protein band of around 100 kD most likely corresponds to PARP1 itself (Fig 3B). Signals of [$^{32}$P]-labelled pADPr polymers disappeared when reactions were conducted in the presence of the PARP1 inhibitors PJ34 or olaparib (Fig 3B). So far, the data indicate that in this experimental system, DNA-bound proteins are poly(ADP-ribosyl)lated, ubiquitylated, and phosphorylated.

It has been reported that poly(ADP-ribosyl)ation accelerates ubiquitin chain formation at laser-induced DNA-damaged sites (Yan et al, 2013). Furthermore, the E3 ligases CHFR and Iduna (also known as RNF146) associate directly with pADPr chains (Oberoi et al, 2010; Kang et al, 2011). To test if PARP1 activity also promotes the recruitment of the E1 ubiquitin-activating enzyme UBA1 to DNA, we incubated the linear duplex DNA structures and protein extracts in the presence of PJ34 and olaparib (Fig 3C). PARP1 inhibitors did not affect the recruitment of UBA1 to DNA, indicating that pADPr chains synthesised during the reaction were not necessary for the association of UBA1 with the biotinylated DNA structures (Fig 3C). This observation, however, did not exclude the possibility that UBA1 binds to pADPr chains pre-existing in the extract before incubation with DNA substrate. Poly(ADP-ribosyl)ation is normally a short-lived protein modification rapidly reversed by poly(ADP-ribose) glycohydrolase (PARG) (Crawford et al, 2018). To evaluate the stability of pADPr polymers in this cell-free system, we performed a time course analysis of [$^{32}$P]-labelled pADPr signals pulled down with the biotinylated gapped linear duplex DNA structure. The level of poly(ADP-ribosyl)ated proteins

was strikingly high after 30-min incubation and declined only after 60–120 min (Fig S1D). To ensure the elimination of pADPr chains, we treated nuclear extracts with PARG. In comparison with control protein extracts, the amount of UBA1 pulled-down with biotinylated DNA was strongly reduced in PARG-treated protein extracts, indicating that pADPr polymers are required for UBA1 recruitment to DNA (Fig 3D). Next, we prepared extracts from PARP1 knockdown cells as an alternative approach to yield extracts free of ADP-ribosylated proteins, including PARP1, which auto–ADP-ribosylates itself extensively (Fig 3E). Suppression of PARP1 did not compromise the capacity of DNA-PKcs to bind DNA, as expected, but inhibited the recruitment of UBA1 to linear duplex DNA structures (Fig 3F). PARP1 depletion completely abrogated Chk1 Ser345 phospho-signals in DNA pull-downs, consistent with the role of PARP1 in the recruitment of Chk1 to ATR-signalling complexes (Min et al, 2013), and of UBA1, which is also required for Chk1 phosphorylation (Fig 2). Collectively, these data suggest that UBA1 is recruited to DNA in a PARP1-dependent manner and that this recruitment is required for full Chk1 phosphorylation. To check if UBA1 recruitment to cellular replication forks is also PARP1 dependent, we performed iPOND experiments in the presence of PJ34 and olaparib (Fig S1E). In vivo, PARP1 inhibitors were sufficient to reduce UBA1 recruitment to forks, without a noticeable impact on PCNA loading, suggesting that PARP1 activity also promotes UBA1 recruitment to forks in vivo. Taken together, the data indicate that pADPr polymers promote UBA1 recruitment to chromatin.

### Human UBA1 binds to pADPr

Because UBA1 is recruited to linear duplex DNA structures in a PARP1-dependent manner, we surmised that UBA1 might directly associate with pADPr chains. However, using the Pfam database, we did not identify in UBA1 any of the pADPr-binding motifs described to date (Teloni & Altmeyer, 2016). To test experimentally if human UBA1 exhibits affinity for pADPr, we dot-blotted purified recombinant UBA1 on a nitrocellulose membrane, incubated the membrane with [$^{32}$P]-labelled pADPr polymers, and then subjected the membrane to stringent Tris buffered saline with Tween (TBST) washing. In comparison with recombinant H2A and BSA proteins, used as positive and negative pADPr-binding controls, respectively, we observed that human UBA1 binds pADPr polymers directly (Fig 4A). We confirmed this result using an anti-pADPr antibody to reveal pADPr polymers associated with UBA1 immobilised on a nitrocellulose membrane (Fig 4B and C). In striking contrast, S. cerevisiae Uba1 and human UBA6 exhibited little affinity for pADPr polymers (Fig 4B and C), consistent with the fact that UBA6 is not implicated in the DDR (Moudry et al, 2012) and that S. cerevisiae is not provided with a poly(ADP-ribosyl)ation system (Perina et al, 2014).

To identify the region in human UBA1 required for pADPr binding, we expressed and purified to near homogeneity six overlapping protein fragments covering the entire UBA1 sequence and fused them to maltose-binding proteins at their amino terminus (Fig 4D). The fragment that includes amino acids 571–800 in human UBA1 exhibited robust affinity for pADPr, as revealed by immunoblot analysis (Fig 4E). To confirm this, we incubated $^{MBP}$UBA1$^{(571–800)}$ with

pADPr polymers and retrieved $^{MBP}$UBA1$^{(571–800)}$ by amylose affinity and maltose elution. Immunoblotting of the eluates spotted on a nitrocellulose membrane confirmed that pADPr associates with $^{MBP}$UBA1$^{(571–800)}$ (Fig 4F). These results indicate that a domain located between amino acids 571 and 800 endows human UBA1 with the capacity to bind to pADPr.

### UBA1 residues conserved in animals (but not in fungi) are critical for pADPr binding

To determine the amino acids required for binding of human UBA1 to pADPr, we decided to use an evolutionary approach. We queried the UniProt database for UBA1 orthologues and retrieved 104 UBA1 sequences that were at least 500 amino acids long (Table S2). We also collected six UBA6 sequences from the Animalia eukaryotic subgroup (Table S2). Next, we aligned all sequences with MAFFT (multiple sequence alignment program for unix-like operating systems) using the L-INS-I algorithm (Katoh & Standley, 2013). The evolutionary tree of UBA1 sequences was reconstructed using maximum likelihood and Bayesian algorithms, treating UBA6 sequences as an out-group and using the entire sequence from each species. The topologies of the trees produced by the different algorithms were almost identical and in good agreement with the species phylogeny (Figs 5A and S2). Different eukaryotic subgroups were nearly perfectly separated, whereas UBA6 sequences formed a distinct cluster, at some distance from UBA1 sequences (Fig 5A).

Human and S. cerevisiae UBA1 exhibit more than 80% sequence similarity overall. We hypothesised that the region required for the binding of human UBA1 to pADPr should not be as conserved in S. cerevisiae UBA1, as the latter does not exhibit affinity for pADPr. We examined a section of the alignment that encompasses amino acids 571–800 in human UBA1, that is, the region shown above to be implicated in pADPr binding. We noticed a 58–amino acid sequence that includes basic and hydrophobic amino acids in human UBA1 that was not conserved in the yeast orthologue nor in human UBA6 (Fig 5B). More specifically, we focused on seven amino acids in this region (Fig 5B): positions 655 (L in human and Y in yeast), 657 (K in human and T in yeast), 671 (K in human and N in yeast), 675 (R in human and Q in yeast), 679 (L in human and Q in yeast), 680 (A in human and S in yeast), and 697 (L in human and S in yeast). These amino acids differ in charge and hydrophobicity and are localised in a solvent-exposed surface area of the UBA1 (Fig 5C). Hence, these residues are, in principle, accessible for binding to the pADPr polyanion. To evaluate their degree of conservation across Animalia and Fungi, we generated logos of the alignment for the entire protein region (Fig 6A), using the command line implementation of WebLogo (Crooks et al, 2004). Stronger sequence conservation could be observed within each phylogenetic subgroup than across them, highlighting a divergence between the two evolutionary lineages.

We used site-directed mutagenesis to test the hypothesis that solvent-exposed basic and hydrophobic residues conserved in Animalia constitute a pADPr polymer-binding surface for UBA1. We produced mutated $^{MBP}$UBA1$^{(571–800)}$ fragments carrying substitutions of the human amino acids into their yeast counterparts. We purified to near homogeneity L655Y-, K657T-, K671N-, K675Q-, L679Q-A680S-, and

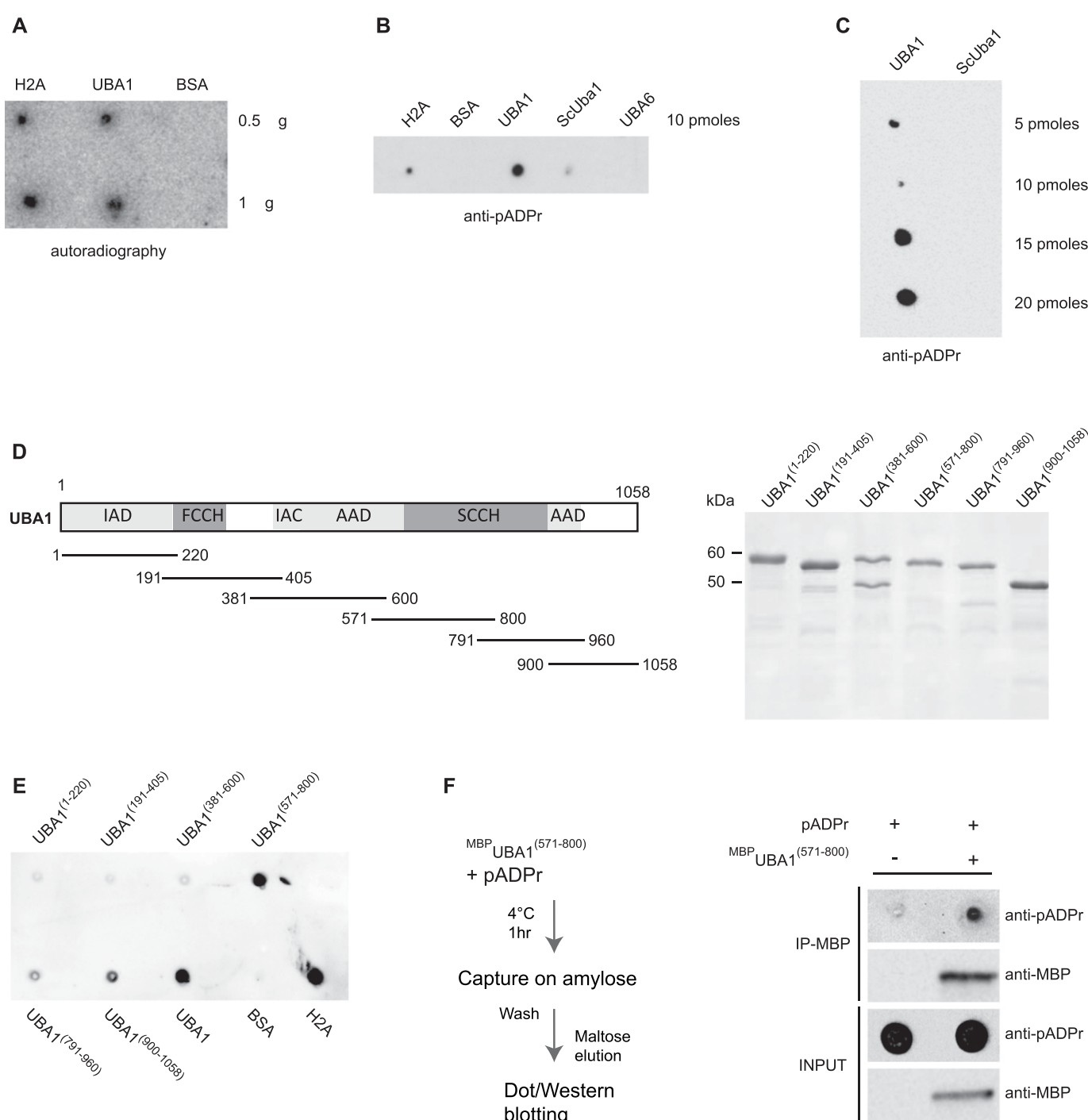

**Figure 4.   Human UBA1 binds to pADPr chains.**
**(A)** *Homo sapiens* UBA1, H2A, and BSA were spotted on a nitrocellulose membrane, incubated with [$^{32}$P]-labelled pADPr, washed, and exposed to autoradiography. **(B)** Purified H2A, BSA, *H. sapiens* UBA1 (UBA1), *S. cerevisiae* Uba1 (ScUba1), and *H. sapiens* UBA6 (UBA6) were spotted on nitrocellulose membrane and incubated with purified pADPr chains. The retention of pADPr on the membrane was revealed using an anti-pADPr antibody. **(C)** Increasing amounts of purified *H. sapiens* UBA1 (UBA1) and *S. cerevisiae* Uba1 (ScUba1) were spotted on nitrocellulose membrane and incubated with purified pADPr chains. The retention of pADPr on the membrane was revealed using an anti-pADPr antibody. **(D)** Left panel: schematic representation of UBA1 with its functional domains and the six purified overlapping fragments of UBA1. Right panel: Purified MBP-UBA1 fragments resolved by PAGE and stained by Coomassie Brilliant Blue. **(E)** UBA1 fragments (10 pM) were spotted on nitrocellulose membrane incubated with purified pADPr chains. Immobilised pADPr was revealed using an anti-pADPr antibody. **(F)** Left panel: experimental scheme. $^{MBP}$UBA1$^{(571–800)}$ was incubated with pADPr polymers, captured on an amylose resin, washed, and eluted with maltose. Eluted $^{MBP}$UBA1$^{(571–800)}$ was resolved by PAGE and revealed via anti-MBP immunoblotting. Eluted pADPr was spotted on a nitrocellulose membrane and revealed using an anti-pADPr antibody. IAD, inactive adenylation domain; FCCH, first catalytic cysteine half domain; AAD, active adenylation domain; SCCH, second catalytic cysteine half domain.

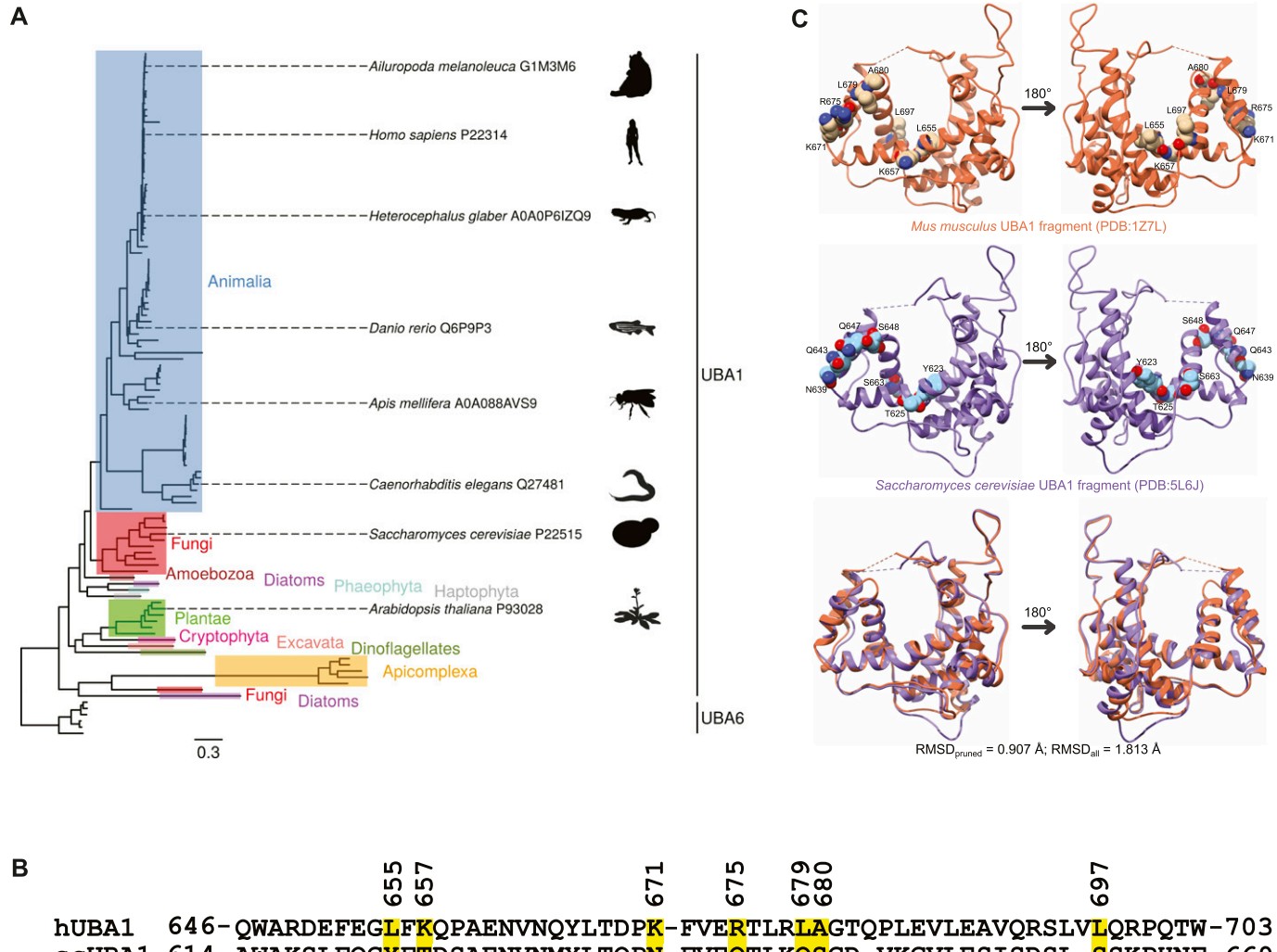

**Figure 5. Analysis of UBA1 and UBA6 protein sequences.**
**(A)** The phylogenetic tree of UBA1 and UBA6 protein sequences, as inferred with MrBayes. The clustering of UBA1 orthologues is in good agreement with the phylogeny of eukaryotic subgroups. Note that evolutionary distance separates UBA1 sequences from the UBA6 out-group. **(B)** Alignment of the region 646–703 from the *H. sapiens* UBA1, and the corresponding regions of ScUba1 and *H. sapiens* UBA6, extracted from a larger alignment of 110 UBA1 and UBA6 sequences. The stars correspond to identical residues. The positions of nonidentical residues that were selected for mutagenesis are highlighted in yellow. **(C)** Structural comparison of a fragment of the *Mus musculus* UBA1 (*H. sapiens* UBA1 crystal structure is not available) and the *S. cerevisiae* orthologue. The seven amino acids that were chosen for mutagenesis are explicitly shown in space-filling representation. The superposition of the two proteins (bottom) highlights their close structural similarity.

L697S- $^{\text{MBP}}$UBA1$^{(571–800)}$ and assessed their capacity to bind pADPr using the nitrocellulose pADPr binding assay (Fig 6B). We dot-blotted increasing concentrations of the mutated protein fragments and detected them with anti-MBP antibody to verify the quality of protein deposition on the membrane (Fig 6B, bottom panel). Immunoblotting with an anti-pADPr antibody revealed that the human-to-yeast amino acid substitutions L655Y, K657T, K671N, and R675Q reduced significantly the affinity of $^{\text{MBP}}$UBA1$^{(571–800)}$ for pADPr polymers (Fig 6B). To confirm the role of these amino acids in pADPr binding, we mixed $^{\text{MBP}}$UBA1$^{(571–800)}$ mutants and pADPr polymers in solution, pulled down $^{\text{MBP}}$UBA1$^{(571–800)}$ with amylose magnetic beads, washed and eluted $^{\text{MBP}}$UBA1$^{(571–800)}$ mutants with maltose, and dot-blotted the eluate on a nitrocellulose membrane to probe pADPr signals with

an anti-pADPr antibody (Fig S3). Human-to-yeast amino acid substitutions in $^{\text{MBP}}$UBA1$^{(571–800)}$ consistently reduced the level of pADPr isolated with $^{\text{MBP}}$UBA1$^{(571–800)}$ (Fig S3). Thus, a single amino acid change was sufficient to reduce the affinity of UBA1 for pADPr chains. These data indicate that a solvent-exposed positively charged patch endows Animalia UBA1 proteins with the capacity to bind polymers of pADPr.

Last, we used an inducible protein replacement system to test if UBA1 binding to pADPr influences replication stress signalling in cultured cells (Ghodgaonkar et al, 2014). We knocked down endogenous UBA1 and expressed shRNA-resistant full-length UBA1 cDNAs encoding amino acid substitutions L655Y, K657T, and K671N (Fig 6C). In comparison with UBA1 knockdown cells

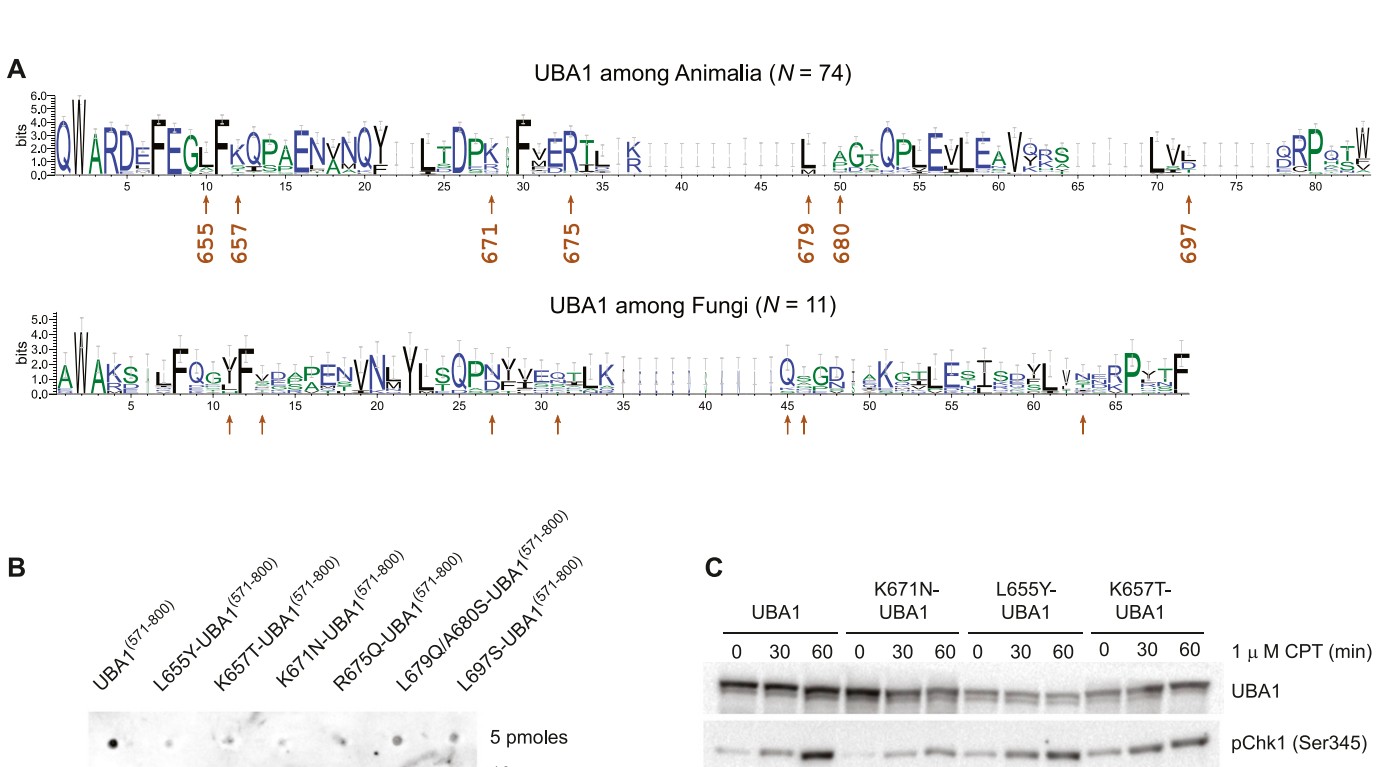

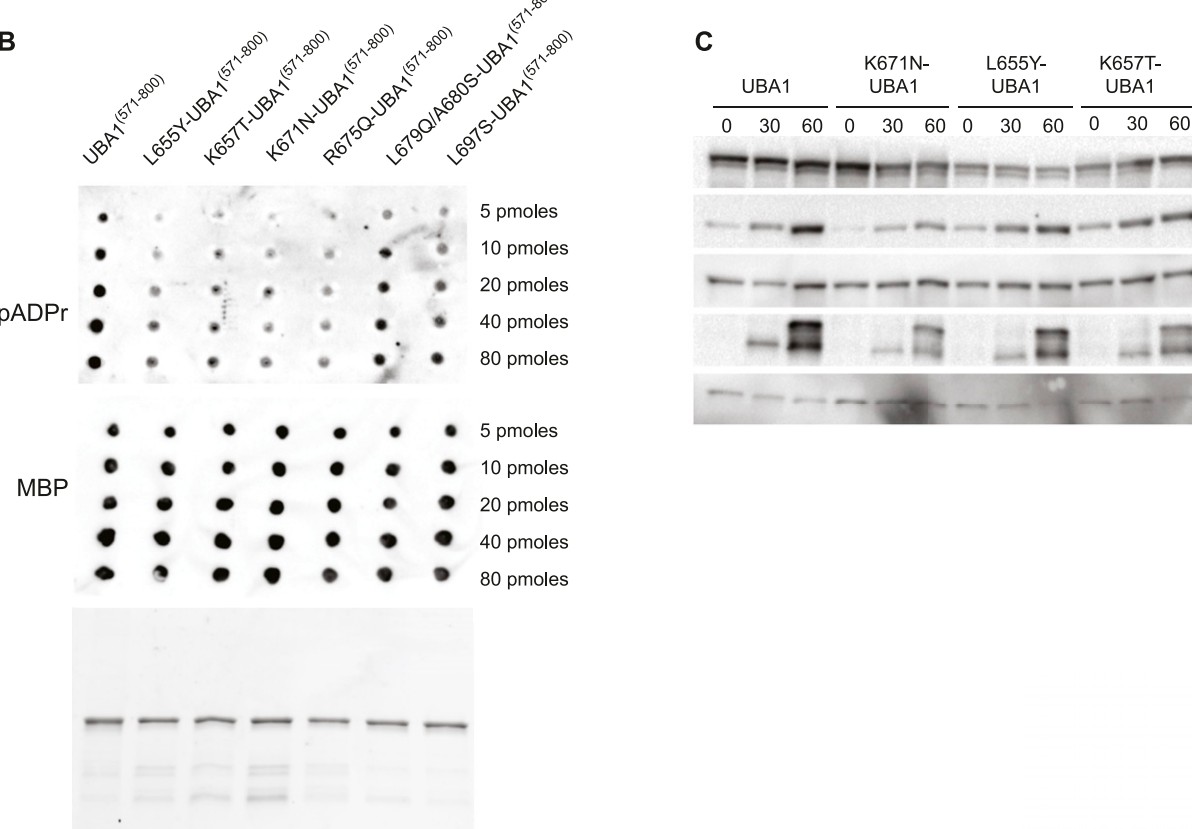

**Figure 6. UBA1 residues conserved in animals (but not in fungi) are critical for pADPr binding.**
**(A)** Sequence logos of the alignment of the region shown in Fig 5B. The height of the letters in each position represents the degree of sequence conservation. Colours are used for differentiation among hydrophilic (blue), hydrophobic (black), or neutral amino acids (green). Error bars correspond to the Bayesian 95% confidence interval of the height. Arrows indicate the positions of the amino acids chosen for mutagenesis. These seven amino acids are not conserved between Animalia and Fungi. **(B)** Increasing amounts (5 to 80 pM) of purified MBPUBA1(571–800) fragments containing the indicated amino acid substitutions were spotted on a nitrocellulose membrane, incubated with purified pADPr chains, and washed. The retention of pADPr was analysed using an anti-pADPr antibody. UBA1 protein fragments used in the pADPr binding assay were revealed using an anti-MBP antibody (middle panel) or resolved by PAGE and stained with Coomassie Brilliant Blue (bottom panel). **(C)** U2OS cells were transfected with pAIO vector encoding an anti-UBA1 shRNA and an shRNA-resistant UBA1 cDNA encoding wild-type or mutated UBA1, as indicated, and treated with doxycyclin for 3 d. Cells were exposed to 1 μM CPT for increasing amounts of time (as indicated). Indicated proteins were detected by Western blotting.

complemented with recombinant wild-type UBA1, the level of Chk1 and RPA32 phosphorylation induced by CPT treatment was reduced in cells expressing pADRr-binding mutant forms of UBA1 (Fig 6C). The data indicate that the binding of UBA1 to pADPr polymers formed in response to DNA replication stress facilitates ATR activation.

# Discussion

Extensive posttranslational modifications of chromatin-associated proteins are required for DNA damage signalling and for the co-ordinated recruitment of DNA repair proteins at DNA damage sites (Ciccia & Elledge, 2010; Dantuma & van Attikum, 2016; Schwertman et al, 2016). Here, we provide insights into the coordination of apical posttranslational modification enzymes PARP1, UBA1, and ATR in the crowded and complex environment of a human protein extract.

We show that UBA1, the E1 enzyme at the apex of the ubiquitylation cascade in the DDR, is recruited to DNA via direct binding to pADPr polymers (Moudry et al, 2012). UBA1 bound indiscriminately to both linear duplex and gapped linear duplex DNA, as both DNA substrates activate poly (ADP-ribosyl)ation. We find that UBA1 recruitment to an ATR-activating DNA structure is required for ATR/Chk1 signalling. We need to emphasise that UBA1 binding to DNA, obviously, is not sufficient for the activation of ATR. The latter strictly depends on the assembly of a combination of sensor and mediator proteins that recognise distinct structural features in DNA, including ssDNA and ssDNA to dsDNA junctions. We identified a solvent-exposed positively charged patch in human UBA1, conserved in the Animalia kingdom, which endows UBA1 with affinity for pADPr chains. The transient association of UBA1 with pADPr may facilitate reiterative ubiquitin activation to sustain the flux of protein modifications at DNA damage sites or stalled replication forks. These data illustrate the utility of human cell-free extracts to dissect the biochemical underpinnings of DDRs. In this work, we studied ATR activation using protein extracts from HeLa S3 cells. Our unpublished experiments indicate that endogenous ATR is also efficiently activated in extracts prepared from different cell lines, including HEK293, U2OS, and multiple myeloma cell lines.

In this study, the transient association of endogenous UBA1 with chromatin in living cells was revealed by iPOND, a procedure that involves formaldehyde cross-linking and isolation of proteins on newly synthesised DNA. This is consistent with the identification by mass spectrometry of UBA1 among proteins localised at DNA breaks induced by UV laser micro-irradiation (Izhar et al, 2015). We show that affinity for pADPr is a specific feature of Animalia UBA1. The second human E1 enzyme UBA6 did not bind to pADPr. This suggests a reason why UBA6 does not overlap with UBA1 in the DDR (Moudry et al, 2012). Neither did S. cerevisiae Uba1 bind to pADPr, consistent with the absence of a pADPr system in the Fungi kingdom (Perina et al, 2014). Although a variety of pADPr-binding motifs have been identified in recent years (Teloni & Altmeyer, 2016), we did not recognise any of these motifs in the primary sequence of human UBA1. The first described pADPr-binding module consists of a loosely defined 20–amino acid region containing hydrophobic amino acids interspaced with basic amino acids (Pleschke et al, 2000). In recent years, a continually increasing variety of protein motifs such as phosphate-binding pockets, oligonucleotide/oligosaccharide-binding folds, RNA recognition motifs, and low-complexity glycine–arginine–rich domains exhibit affinity for pADPr polymers and have been categorised as "pADPr readers" (Teloni & Altmeyer, 2016). Proteins that do include one of the abovementioned motifs, however, do not necessarily bind to pADPr. Reciprocally, some proteins bind to pADPr but do not include any recognisable pADPr-binding motifs. One

example is Chk1 that is fully activated upon binding to pADPr near replication forks (Min et al, 2013).

Using a biochemical and an evolutionary approach, we identified key amino acids responsible for human UBA1 binding to pADPr. Whereas the sequence of amino acids involved in human UBA1 binding to pADPr adopts a similar fold in mouse (similar to human) and in yeast UBA1, we identified one hydrophobic amino acid and three basic amino acids in human UBA1 that are not conserved in fungi and that are required for binding to pADPr polyanions. The amino acids are located in a solvent-accessible surface of UBA1. We conclude that the electrostatic surface potential of UBA1 rather than a specific fold explains the affinity of human but not yeast UBA1 for pADPr polymers. Extensive work over the past two decades illustrates that the affinity of a protein for pADPr is hardly predictable in silico. We propose that a solvent-exposed positively charged surface is sufficient for a protein to have affinity for pADPr polymers.

It has been proposed that the anionic pADPr scaffold may contribute to the organisation of cellular architectures (Leung, 2014). Poly(ADP-ribosyl)ation triggers nuclear re-localisation of RNA-binding proteins in response to genotoxic stress (Jungmichel et al, 2013; Izhar et al, 2015). The pADPr scaffold may represent a highly dynamic seeding platform for protein recruitment by noncovalent interactions (Leung, 2014). Proteins that are rapidly and transiently recruited to the pADPr scaffold at DNA damage sites include the ATR effector kinase Chk1 (Min et al, 2013), as well as DNA repair, chromatin remodelling, and RNA biogenesis factors (Gagne et al, 2012; Britton et al, 2014; Izhar et al, 2015). A subset of proteins recruited to pADPr chains can undergo liquid-to-liquid phase separation (Kato et al, 2012; Patel et al, 2015). Indeed, PARP1 activity increases the local concentration of proteins such as FUS, which can self-assemble into liquid droplets under physiological conditions (Kato et al, 2012; Altmeyer et al, 2015; Patel et al, 2015). Intriguingly, we observed that UBA1 inhibition by PYR41 induces the accumulation of DNA-bound Chk1 in an inactive state. The coupling of UBA1 recruitment to PARP1 activity at DNA damage sites could both promote dedicated high-affinity protein interactions and prevent protein aggregation, thereby maintaining protein complexes in a functional state. pADPr can also recruit and regulate E3 ligases, including RNF146/Iduna (Kang et al, 2011; Zhang et al, 2011) and UHRF1 (De Vos et al, 2014), suggesting that specific ubiquitylation cascades may be coupled with PARP1 activity. Likewise, we reported previously that DNA-PKcs can prime ATR activation (Vidal-Eychenie et al, 2013). We found that in this ATR activation system, RNA-processing factors such as FUS and HNRNPUL1 were prominent among DNA-bound proteins. Phosphorylation of FUS by DNA-PKcs prevents fused in liposarcoma (FUS) aggregation and associated toxicity (Monahan et al, 2017). DNA-PKcs may promote ATR activation through phosphorylation of proteins that can undergo phase separation and, thereby, protect against aggregation of pADPr-seeded protein assemblies. This in turn would ensure that high-affinity dedicated interactions required for ATR activation can take place. Given the existence of an abundant PARP1-DNA-PKcs heterodimer in cells (Spagnolo et al, 2012), and the recruitment of UBA1 to pADPr shown here, we propose that the seeding of protein assemblies by PARP1 activity is directly coupled with regulation of phase transitions by phosphorylation and ubiquitylation (Fig 7). Here, we provide evidence that UBA1 recruitment to

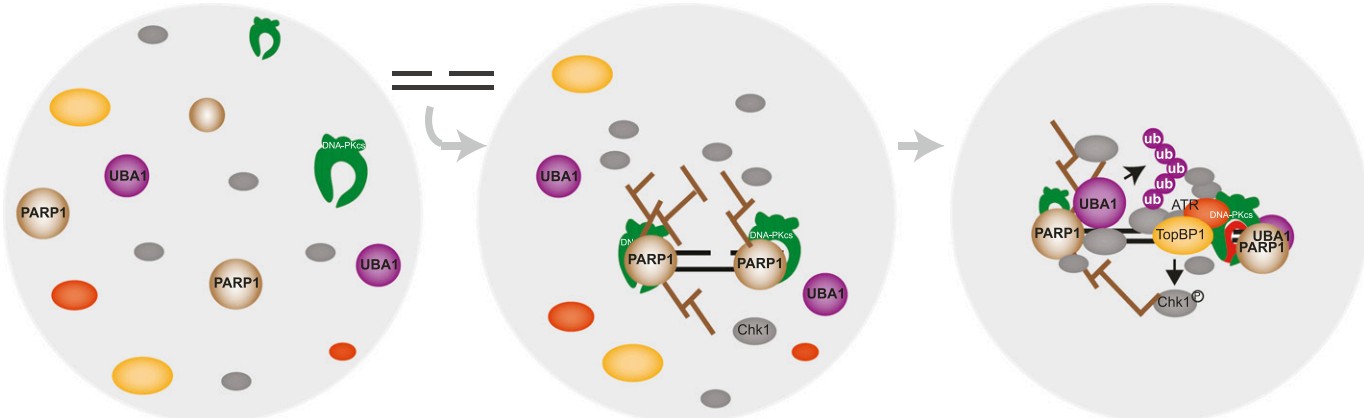

**Figure 7. Model for the formation of ATR-signalling bodies.**
Upon addition of a gapped DNA substrate in human cell-free extracts, poly(ADP-ribosyl)ation accelerates the recruitment of DNA damage signalling proteins, including the ubiquitin-activating enzyme UBA1. The coupling of pADPr-seeded protein assembly with protein ubiquitylation and protein phosphorylation prevents the formation of nonfunctional protein aggregates and promotes dedicated protein–protein interactions required for ATR signalling.

pADPr polymers facilitates ATR signalling in protein extracts. PARP1 inhibition has numerous consequences in cell physiology, including PARP1-mediated Chk1 activation at stalled replication forks (Min et al, 2013), PARP1—regulated replication fork dynamics and repair (Bryant et al, 2009; Ray Chaudhuri et al, 2012; Zellweger et al, 2015), and PARP1-mediated UBA1 recruitment to DNA (this study). To fully explore the functional implications of UBA1 binding to pADPr polymers in cells, a genome editing approach will be necessary to block precisely, using amino acid substitutions, the recruitment of UBA1 to pADPr in experimental conditions that preserve ubiquitin activation by UBA1 and poly(ADP-ribosyl)ation by PARP1.

# Materials and Methods

### Cell lines

HeLa S3 (American Type Culture Collection), HEK293 (American Type Culture Collection), and U2OS T-REx (Thermo Fisher Scientific) cells were cultured in DMEM or Roswell Park Memorial Institute medium. The culture medium was supplemented with 10% foetal bovine serum (Biowest) and penicillin/streptomycin (Sigma-Aldrich). Cells were incubated in 5% $CO_2$ at 37°C. To engineer the UBA1-FLAG cell line, UBA1 was cloned into Met-FLAG-pCR3 vector using the Gateway technology and transfected into the HEK293 cell line.

### Gene silencing

For UBA1 depletion, siRNA oligonucleotides were purchased from Ambion (S600) and transfected using jetPrime (polypus transfection). Anti-UBA1 shRNA (5′-TCCAACTTCTCCGACTAC-3′) was cloned in pAIO, shRNAs against PARP1 (5′-GGGCAAGCACAGTGTCAAA-3′), or luciferase (5′-CTTACGCTGAGTACTTCGA-3′) were cloned in pSUPER-Puro and transfected using lipofectamine (Thermo Fisher Scientific). After 48 h, the cells were lysed and nuclear extracts were prepared as described below.

### Western blot

The proteins were resolved by SDS–PAGE using homemade or precast gels (Bio-Rad) and transferred to a nitrocellulose membrane. Antibodies against the following proteins were used: Ser345 phospho-Chk1 (#2348; Cell Signaling Technology), Chk1 (#sc-8408; Santa Cruz), PCNA (#P8825; Sigma-Aldrich), Ser33 phospho-RPA32 (#ab221887; Abcam), RPA32 (#NA18; Calbiochem), DNA-PKcs (#ab1532; Abcam), PARP1 (#sc-8007; Santa Cruz), UBA1 (#4891s; Cell Signaling or #sc-53555; Santa Cruz), TOPBP1 (#A300-111A; Bethyl Laboratories), ATR (#A300-137A; Bethyl Laboratories), Thr1989 phospho-ATR (#GTX128145; GeneTex), and MBP (#E80329; New England Biolabs).

### Nuclear extract preparation

Nuclear extracts were prepared using Dignam's method as described previously (Shiotani & Zou, 2009; Vidal-Eychenie et al, 2013). Cells were grown to 70–80% confluence, collected by scrapping, centrifuged at 200 $g$ for 3 min at 4°C, and washed twice in PBS. The cell pellet was resuspended into 5× pellet volume of hypotonic buffer A (10 mM Hepes-KOH, pH 7.9, 10 mM KCl, 1.5 mM $MgCl_2$, 0.5 mM DTT, and 0.5 mM PMSF) supplemented with protease inhibitor cocktail and phosphatase inhibitor and kept on ice for 5 min and then centrifuged for 4 min at 200 $g$. The pellet was resuspended into 2× cell volume of buffer A and cells were lysed by Dounce homogenisation using a tight-fitting pestle. Homogenised cells were centrifuged at 4,000 $g$ for 5 min at 4°C. Pelleted nuclei were resuspended into a packed cell volume of buffer C (20 mM Hepes-KOH, pH 7.9, 600 mM KCl, 1.5 mM $MgCl_2$, 0.2 mM EDTA, 25% glycerol, 0.5 mM DTT, and 0.5 mM PMSF) supplemented with a cocktail of protease inhibitor and phosphatase inhibitor and kept rotating for 30 min at 4°C. The supernatant was recovered by centrifugation at 16,000 $g$ for 15 min at 4°C and dialysed against buffer D (20 mM Hepes-KOH, pH 7.9, 10 mM KCl, 0.5 mM DTT, and 0.5 mM PMSF). Dialysed nuclear extract was centrifuged at 100,000 $g$ for 30 min at 4°C to remove residual precipitates. Protein concentration was

determined using Bradford's protein estimation method. Nuclear extract was aliquoted and stored at −80°C until further use.

## Preparation of DNA substrate for in vitro assay

DNA substrates used in this study were generated as described by Vidal-Eychenie et al (2013). Briefly, biotinylated duplex DNA was generated by PCR amplification of plasmid pG68 (Ralf et al, 2006) with the following primers (5′-Biotin-TGCGGCATCAGAGCAGATTG-3′ and 5′-GCACCCCAGGCTTTACACTTTATG-3′). The 573-bp amplified duplex was gapped by digestion with the nicking restriction enzyme NbBbvC1 and heat denaturation as described (Ralf et al, 2006). Gapped DNA is refractory to digestion by SpeI restriction enzyme.

## DNA pull-down assay

In vitro DNA pull-down assay was carried out as explained previously (Vidal-Eychenie et al, 2013). Biotinylated DNA substrates were conjugated to 50 $\mu$l streptavidin-coated magnetic beads (Ademtech) in binding and washing buffer and incubated for 10 min at room temperature. DNA–bead complexes were washed in wash buffer (10 mM Hepes, 100 mM KOAc, and 0.1 mM MgOAc) and resuspended into 50 $\mu$l reaction buffer. Nuclear extract (20 $\mu$g) was added to streptavidin-bound DNA substrates and reactions were incubated for 10 min at 20°C. DNA-bound proteins were washed extensively and resuspended in 20 $\mu$l Laemmli buffer. To reveal ubiquitinated proteins, we supplemented in vitro reaction with 2 $\mu$g of FLAG-ubiquitin (Sigma-Aldrich) and resolved the proteins on precast 4–15% gradient gel. Ubiquitylated proteins were detected with an anti-FLAG antibody (#F7425; Sigma-Aldrich). To visualise poly(ADP-ribosyl)ated proteins during the course of the reaction, we supplemented reaction mixtures with [$^{32}$P]-labelled nicotinamide adenine dinucleotide and performed DNA pull-downs as indicated above. Proteins were resolved on 4–15% gradient gel and visualised by autoradiography. To assess the impact of PARP1 inhibitors on poly(ADP-ribosyl)ation, UBA1 recruitment, and ubiquitylation, we supplemented reaction mixtures with 200 $\mu$M of PJ34 (#sc-204161A; Santa Cruz) or pre-incubated the reaction mixtures for 60 min with 100 $\mu$M olaparib (#S1060; Euromedex) before incubation with the DNA substrates. To test if pADPr chain present on proteins before in vitro reaction were required for UBA1 recruitment, we supplemented the nuclear extract with purified PARG, a kind gift from Jean-Christophe Amé and Valérie Schreiber. To inhibit UBA1 activity, we used PYR41 at 100 $\mu$M (#662105; Calbiochem).

## iPOND

iPOND was performed as described by Lossaint et al (2013) and Ribeyre et al (2016). For mass spectrometry experiments, HeLa S3 cells were pulse-labelled with 10 $\mu$M EdU for 5 min and a 120-min chase was performed with 10 $\mu$M thymidine. The cells were fixed with 1% formaldehyde for 5 min followed or not by quenching of formaldehyde by 5-min incubation with 0.125 M glycine. Fixed samples were collected by centrifugation at 2,000 rpm for 3 min, washed three times with PBS, and stored at −80°C. The cells were permeabilised with 0.5% triton and click chemistry was used to conjugate biotin to EdU-labelled DNA. The cells were resuspended in lysis buffer and sonication was performed using a Qsonica sonicator. Biotin-conjugated DNA–protein complexes were captured using streptavidin beads (Ademtech). Captured complexes were washed with lysis buffer and high salt containing solution. Proteins associated with nascent DNA were eluted under reducing conditions by boiling into SDS sample buffer for 30 min at 95°C. To confirm the presence of UBA1 on nascent DNA, we repeated the experimental procedure using HEK293 cells that support high-yield iPOND and are, therefore, traditionally used in iPOND experiments. For Western blot experiments, HEK293 cells were labelled with 10 $\mu$M of EdU for 10 min (when indicated, cells were pretreated for 120 min with 10 $\mu$M PJ34 or 10 $\mu$M olaparib). A 60-min chase with 10 $\mu$M thymidine was performed when indicated.

## Mass spectrometry analysis

Protein extracts were resolved by SDS–PAGE (10%) and detected with Coomassie Brilliant Blue staining. Electrophoretic lanes were cut in five fractions and the gel pieces were washed with water, dehydrated using 50% acetonitrile (ACN) in 50 mM NH$_4$HCO$_3$ and then 100% ACN, and dried. After DDT reduction and iodoacetamide alkylation, the gel pieces were re-swollen in a 0.1 $\mu$g/$\mu$l trypsin (Promega) solution (100 mM NH$_4$HCO$_3$, 0.5 M CaCl$_2$, and 1% ProteaseMax). Resulting peptides were trapped and desalted on C18 Zip-Tips (Agilent) and speed vacuum concentrated. For liquid chromatography tandem-mass spectrometry, the peptide mixtures were dissolved in 10 $\mu$l 0.1% formic acid (FA) and loaded on an Ekspert 425 nanoLC system (SCIEX) equipped with a C18 column (Discovery BIO Wide Pore, 3 $\mu$m, 0.5 × 10 cm; Supelco). The mobile phases were solvent A (water and 0.1% FA) and B (ACN and 0.1% FA). Injection was performed with 95% solvent A at a flow rate of 5 $\mu$l/min. The peptides were separated with the following gradient: 5–40% B in 100 min, 40–80% B in 5 min, and the separation was monitored online on a TripleTOF 5600 mass spectrometer (SCIEX). The total ion chromatogram acquisition was made in information-dependent acquisition mode using Analyst TF v.1.6 software (SCIEX). Each cycle consisted of a time-of-flight mass spectrometry (TOF-MS) spectrum acquisition for 250 ms (350–1,600 kD), followed by acquisition of up to 30 MS/MS spectra (75 ms each, mass range 100–1,600 kD) and of MS peaks above intensity 400 taking 2.5 s total full cycle. Target ions were excluded from the scan for 15 s after being detected. The information-dependent acquisition advanced rolling collision energy option was used to automatically ramp up the collision energy value in the collision cell as the $m/z$ value was increased. Protein identification was performed by the ProteinPilot software v.5.0 (SCIEX). From each MS2 spectrum, the Paragon algorithm was used to search UniProt/Swiss-Prot database (release 2015) with the following parameters: trypsin specificity, $cys$-carbamidomethylation, and search effort set to rapid. After database searching, only proteins identified with an unused score of 2 and peptides identified with a confidence score of 95 were retained. Analysis of raw files was performed using MaxQuant version 1.5.6.5 using default settings with label-free quantification option enabled. Raw file spectra were searched against the human UniProt reference database. Protein, peptide, and site false discovery rate were adjusted to <0.01.

**Purified proteins**

Purified human UBA1 (E-305), purified *S. cerevisiae* Uba1 (E-300), and human UBA6 (E-307) were acquired from Boston Biochem. Purified histone H2A (#M2502S) was obtained from New England Biolabs.

**Radioactive pADPr binding assay**

Radioactive pADPr binding assay was performed as described by Ahel et al (2008), with some modifications. Briefly, proteins were spotted on nitrocellulose membrane and allowed to dry for 15 min and subsequently blocked in 1× TBS-T buffer supplemented with 5% milk. The membrane was incubated with radioactive pADPr for 30 min, washed extensively (three washes with TBS-T and three times with TBS-T containing 1 M NaCl), air-dried, and subjected to autoradiography. Radiolabelled pADPr was prepared by incubating 50 U of auto-modified PARP1 enzyme using Trevigen PARP activity assay kit. pADPr chains were detached from PARP1 by treating them with DNAse1 for 1 h and proteinase K for 2 h. Water-soluble pADPr chains were extracted using phenol–chloroform extraction and diluted in 10 ml TBS-T.

**Nonradioactive pADPr binding assay**

Nonradioactive pADPr binding assay was performed as described by Britton et al (2014), with some modifications. We spotted purified proteins on a nitrocellulose membrane before incubation for 1 h with 10 nM of pADPr chains at room temperature. After extensive washing, the membrane was incubated overnight at 4°C with an anti-pADPr antibody (#4335-MC-100; Trevigen). Membranes were washed extensively with TBST and incubated with a mouse secondary antibody coupled with HRP and then revealed using ECL method with Bio-Rad Gel Doc system.

**Construction of UBA1 expression plasmids**

Six different overlapping UBA1 fragments were amplified from UBA1 cDNA (a gift from Dimitris Xirodimas) using KOD Hot start enzyme (Invitrogen) with primers containing *Sal*I and *BamH*I recognition sites. PCR products were cloned into *Sal*I and *BamH*I sites of the pMAL-C5X vector (New England Biolabs).

**Expression and purification of UBA1 proteins from *Escherichia coli***

Maltose-binding protein (MBP)-UBA1 fragments (100 nM of each) were incubated with 100 nM pADPr chains for 1 h at 4°C. When OD reached 0.6, the cells were induced with 0.5 mM IPTG and 0.2% L-arabinose for 2 h. The cells were harvested by centrifugation at 5,000 *g*, resuspended in amylose buffer (20 mM Tris, pH 7.4, NaCl 200 mM, DTT 1 mM, and 1× protease inhibitor), and stored at −20°C. The cells were thawed in ice-cold water and lysed by sonication. Soluble fractions were collected by centrifugation at 9,000 *g* for 30 min. The supernatant was loaded on pre-equilibrated amylose column (New England Biolabs) at 4°C. The column was washed with 10 column volumes of amylose buffer. The protein was eluted using amylose buffer supplemented with 10 mM maltose. MBP-UBA1 containing fractions were then fractionated on a Superdex 16/60 column. Purified MBP-UBA1

preparations were analysed by 10% PAGE and Coomassie Brilliant Blue staining and aliquots were stored at −80°C.

**Isolation of MBP-UBA1–pADPr complexes using amylose**

MBP-UBA1 fragments (100 nM of each) were incubated with 100 nM pADPr chains for 1 h at 4°C. Amylose magnetic beads (New England Biolabs) were added and incubated for approximately 30 min and then washed five times with washing buffer (10 mM Hepes, 100 mM KOAc, and 0.1 mM MgOAc). Proteins were eluted with 10 mM maltose and spotted on nitrocellulose membrane to detect pADPr polymers as described above or resolved on 10% SDS–PAGE gel to detect MBP-UBA1 proteins.

**Phylogenetic tree reconstruction**

The quality of the alignment that MAFFT produced was evaluated with ZORRO (Wu et al, 2012), which assigned a quality score between 0 and 10 to each column. In this case, we removed all columns with a score of 0.4 or lower. From this reduced alignment, phylogenetic trees were reconstructed with RAxML v. 8.2.10 (Stamatakis, 2014), PhyML v. 20151210 (Guindon et al, 2010), and MrBayes v. 3.2.6 (Ronquist et al, 2012). The best fitting model of amino acid substitution was identified with RAxML according to the corrected Akaike information criterion (Hurvich & Tsai, 1989), and was subsequently used for tree estimation by all three programs. In particular, we reconstructed 100 distinct trees with RAxML and accepted the one with the highest log-likelihood value. For MrBayes, we executed four independent runs with seven Metropolis-coupled Markov chains per run, for a total of 9,000,000 generations. Trees and parameter estimates were sampled from the posterior distribution every 1,000 generations, after discarding the first 25% of generations as burn-in. To verify that the four independent runs had converged on the same result, we performed several diagnostic tests using the rwty R package v. 1.0.1 (Warren et al, 2017), including split frequency comparisons and visualisation of the tree space that each run explored (Fig S2A–D). We obtained the final MrBayes tree by computing the extended majority-rule consensus tree (i.e., a fully bifurcating topology). The statistical support for each node of the trees that RAxML and PhyML produced was evaluated by bootstrapping. More precisely, we performed 100 nonparametric bootstraps with PhyML, whereas for RAxML, the number of necessary bootstrap replicates was identified according to the extended majority-rule consensus tree criterion (Pattengale et al, 2010). Finally, we compared the three resulting tree topologies by calculating the Matching Split distances (Bogdanowicz & Giaro, 2012a) among them using the TreeCmp program v. 1.0-b291 (Bogdanowicz et al, 2012b) (Fig S2E and F).

**Structural comparison**

To structurally compare the human and yeast UBA1 orthologues and obtain an understanding of the location of the seven amino acids that were chosen for mutagenesis, we queried the Protein Data Bank for experimentally determined structures. Although the structure of the *S. cerevisiae* orthologue was available (PDB: 5L6J), only a fragment of the structure of the human UBA1 was solved (PDB: 4P22) and did not include the amino acids of interest. To circumvent this limitation, we used the *Mus musculus* UBA1

structure (PDB: 1Z7L) instead as (i) its sequence is almost completely identical to the human sequence (96% sequence identity) and (ii) the seven amino acids were conserved between the two proteins. The two structures were superimposed with the UCSF Chimera's MatchMaker tool (v. 1.11.2; Pettersen et al, 2004), using default options. Briefly, MatchMaker repeatedly aligns the two structures, excluding pairs of residues that are more than 2 Å apart at each iteration. As the murine structure comprised only the second catalytic cysteine half domain of UBA1—in which the amino acids of interest are located—and not the entire protein, we also manually removed all nonmatching amino acids of the yeast structure, for comparison and visualisation purposes.

### Generation of UBA1 point mutants

The different point mutations in UBA1$^{(571–800)}$ were produced using Stratagene QuickChange site-directed mutagenesis kit. Primers for site-directed mutagenesis were designed using QuickChange primer designing program: (http://www.genomics.agilent.com/primerDesignProgram.jsp). C5X pMAL vector containing UBA1 F4 DNA insert was used as the template for amplification. Briefly, 25 $\mu$l amplification reaction consisted of 2.5 $\mu$l of 10× QuickChange multi-reaction buffer, 0.75 $\mu$l QuickSolution, 1 $\mu$l dsDNA template (100 ng), 1 $\mu$l primer (100 ng), 1 $\mu$l dNTPs mix, and 1 $\mu$l multienzyme blend. PCR amplification was carried out with an initial denaturation at 95°C for 1 min followed by 30 cycles of denaturation at 95°C for 1 min, annealing at 55°C for 1 min, and extension at 65°C for 2 min/kb. 10 U of DpnI restriction enzyme was added to each amplification reaction and incubated for 1–2 h at 37°C to digest parental DNA. 5 $\mu$l of DpnI-treated DNA was then transformed into XL-10 Gold ultra-competent cells.

### Protein replacement system

To study the impact of UBA1 point mutants on checkpoint activation in vivo, we took advantage of an inducible replacement system (Ghodgaonkar et al, 2014). We first inserted an shRNA directed against UBA1 (5'-TCCAACTTCTCCGACTAC-3') using annealed oligonucleotide 1 (5'-GATCCCCTCCAACTTCTCCGACTACATTCAAGAGATGT-AGTCGGAGAAGTTGGATTTTTGGAAA-3') and oligonucleotide 2 (5'-AGCTTTTCCAAAAATCCAACTTCTCCGACTACATCTCTTGAATGTAGTCG-GAGAAGTTGGAGGG-3'), and HindIII and BglII restriction sites in pAIO vector (a gift from Josef Jiricny). Human UBA1 cDNA (a gift from Dimitris Xirodimas) was modified by site-directed mutagenesis to make it resistant to the shRNA and inserted in pAIO vector containing anti-UBA1 shRNA using BamHI and EcoRV restriction sites. UBA1 point mutations (K671N, L665Y, and K657T) were introduced by site-directed mutagenesis. Plasmids were transfected in U2OS T-Rex cells (a gift from Sébastien Britton) and stable cell lines were obtained via puromycin selection. Induction of shRNA against UBA1 and UBA1 cDNA was triggered using 10 $\mu$g/ml doxycycline for 3 d.

## Methodology

All the experiments described in this article have been repeated at least two times.

### Data availability

All relevant data are available from the authors.

## Supplementary Information

## Acknowledgements

We thank Patrick Calsou, Dimitris Xirodimas, Françoise Dantzer, Gwenaël Rabut, and lab members and alumni for their precious help on the project. We thank Jean-Christophe Amé and Valérie Schreiber for purified PARG, Josef Jiricny for plasmid (all-in-one) vector, and Sébastien Britton for U2OS T-REx cells. We acknowledge the support of the Site de Recherche Intégrée sur le Cancer Montpellier Cancer grant INCa_INSERM_DGOS_12553. This work was supported by grants from la Fondation ARC pour la Recherche sur le Cancer (PGA1 RF20180206787) and Merck Sharp and Dohme Avenir (GnoSTic) to A Constantinou. Ramhari Kumbhar benefitted from a 3-year PhD fellowship from the Labex EpiGenMed, an "investissements d'avenir" program (reference ANR-10-LABX-12-01), and a fourth-year PhD fellowship from la Fondation pour la Recherche Médicale.

### Author Contributions

R Kumbhar: conceptualisation, data curation, formal analysis, funding acquisition, investigation, visualisation, methodology, and writing—original draft.
S Vidal-Eychenié: data curation, investigation, visualisation, and methodology.
DG Kontopoulos: conceptualisation, data curation, formal analysis, investigation, visualisation, and methodology, and writing—original draft.
M Larroque: data curation, formal analysis, investigation, and methodology.
C Larroque: data curation, formal analysis, and methodology.
J Basbous: conceptualisation and methodology.
S Kossida: conceptualisation, data curation, formal analysis, supervision, validation, methodology, and writing—original draft.
C Ribeyre: conceptualisation, data curation, formal analysis, supervision, validation, investigation, visualisation, methodology, and writing—original draft.
A Constantinou: conceptualisation, data curation, supervision, funding acquisition, validation, visualisation, methodology, project administration, and writing—original draft, review, and editing.

### Conflict of Interest Statement

The authors declare that they have no conflict of interest.

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
