## [Reviewer comments · Life Science Alliance]

Recruitment of Ubiquitin-activating enzyme UBA1 to DNA by Poly(ADP-ribose) promotes ATR signalling

Ramhari Kumbhar, Sophie Vidal-Eychenié, Dimitrios Georgios Kontopoulos, Marion Larroque, Christian Larroque, Jihane Basbous¹, Sofia Kossida, Cyril Ribeyre and Angelos Constantinou
DOI: 10.26508/lsa.201800096

Review timeline:

Submission Date:	29 May 2018
Editorial Decision:	31 May 2018
Revision Received:	14 June 2018
Accepted:	14 June 2018

Report:

(Note: Letters and reports are not edited. The original formatting of letters and referee reports may not be reflected in this compilation.)

Please note that the manuscript was previously reviewed at another journal and the reports were taken into account in the decision-making process at Life Science Alliance. Since the original reviews are not subject to Life Science Alliance's transparent review process policy, the reports and author response cannot be published.

1st Editorial Decision

31 May 2018

Thank you for submitting your revised manuscript entitled "Recruitment of Ubiquitin-activating enzyme UBA1 to DNA by Poly(ADP-ribose) promotes ATR signalling" to Life Science Alliance. Your manuscript was reviewed and revised for another journal before, and you provided the reviewer reports of both rounds of peer review to us.

The reviewers were concerned that the revised manuscript does not provide sufficient insight into the physiological significance of your findings. This is in our view not a concern for publication in Life Science Alliance, and we would be happy to publish your work pending final minor revisions to clarify some lingering concerns of the reviewers and to meet our formatting guidelines. No further experiments need to be performed.

I would thus like to ask you to upload a final version of your manuscript (see formatting guidelines below). Please provide a point-by-point response to the remaining concerns of the reviewers and amend your manuscript text/figures accordingly. As mentioned above, no additional experiments are needed, but the issue of additional factors being recruited to the gapped versus duplex oligos, and whether some of those are contributing separately to Chk1 activation seems to have caused confusion (reviewer #1) and should be addressed in the point-by-point response and discussed more explicitly in the text in our view. The other issues mentioned by this reviewer should also be addressed in the point-by-point response and by re-wording the text. Reviewer #2's concern can be addressed as already outlined in your cover letter (discussing that binding is likely not taking place via a 'motif'). Reviewer #3's concern regarding use of only HeLa cells should be discussed as a limitation, and this reviewer's request for more PARP inhibition analyses should be discussed in the text as well. Please also address the minor points regarding text/figure representation of this reviewer.

A. FINAL FILES:

-- High-resolution figure, supplementary figure and video files uploaded as individual files: See our detailed guidelines for preparing your production-ready images, <http://life-science-alliance.org/authorguide>

B. MANUSCRIPT ORGANIZATION AND FORMATTING:

Full guidelines are available on our Instructions for Authors page, <http://life-science-alliance.org/authorguide>

Thank you for your attention to these final processing requirements.

Thank you for submitting your revised manuscript entitled "Recruitment of Ubiquitin-activating enzyme UBA1 to DNA by Poly(ADP-ribose) promotes ATR signalling". We appreciate that you provided a point-by-point response to the concerns raised after re-review at a different journal and the changes made to the manuscript. It is a pleasure to let you know that your manuscript is now accepted for publication in Life Science Alliance. Congratulations on this interesting work.

DISTRIBUTION OF MATERIALS:

Again, congratulations on a very nice paper. I hope you found the process to be constructive and are pleased with how the manuscript was handled editorially. We look forward to future exciting submissions from your lab.